# Synergistic Effects of a Probiotic Culture Extract and Antimicrobial Combinations against Multidrug-Resistant *Acinetobacter baumannii*

**DOI:** 10.3390/medicina59050947

**Published:** 2023-05-15

**Authors:** Ji Hyeon Lee, Joon Kim, Ga-Yeon Kim

**Affiliations:** 1Department of Clinical Laboratory Science, Graduate School, Dankook University, 119 Dandae-ro, Dongnan-Gu, Cheonan-Si 31116, Republic of Korea; 2Department of Laboratory Medicine, Ajou University Hospital, 164 World Cup-ro, Yeongtong-Gu, Suwon-Si 16499, Republic of Korea; 3Department of Public Health, Graduate School, Dankook University, 119 Dandae-ro, Dongnan-Gu, Cheonan-Si 31116, Republic of Korea

**Keywords:** multidrug-resistant *Acinetobacter baumannii*, probiotics, checkerboard, time kill assay, synergistic effects

## Abstract

*Background and Objectives*: Developing effective treatment outcomes for multidrug-resistant *Acinetobacter baumannii* (MRAB) infections, with colistin and tigecycline the current frontline therapies, is challenging, because of the risk of renal toxicity and low blood concentrations of active ingredients when administered intravenously. This study aimed to identify the effect of combination therapy using conventional antimicrobial agents that are used for treating drug-resistant bacteria and the additional synergistic effect of four probiotic culture extracts isolated from the human body and *Lactobacillus* preparations. *Materials and Methods*: The antimicrobial combination and synergistic effect of adding *Lactobacillus* extract against 33 strains of *A. baumannii* isolated from pus, urine, and other specimens submitted to the Department of Laboratory Medicine of a university hospital, located in Gyeonggi-do, Korea, was investigated over a 3-year period between January 2017 and December 2019. *Results*: Antimicrobial susceptibility tests on bacteria isolated in clinical practice demonstrated that 26 strains (79%) were MRAB, while multi-locus sequence typing indicated that ST191 was the predominant type (45%; *n* = 15). Checkerboard test results demonstrated that combination therapy using meropenem and colistin had the highest synergistic effect (fractional inhibitory concentration index = 0.5), while the time–kill assay test using *Lactobacillus* spp. culture extract exhibited an inhibitory effect within 1 h and complete inhibition of MRAB within 3 h. *Lactobacillus paracasei* exhibited the fastest antimicrobial reactivity and longest sustained antimicrobial activity. *Conclusion*: These findings provide useful foundational data for an appropriate combination of colistin with other antimicrobial agents for treating MRAB infection in clinical settings, and the use of various probiotic culture extracts to reduce the required dosage, and therefore toxicity of colistin.

## 1. Introduction

Since the discovery of penicillin by Sir Alexander Fleming, antibacterial agents have become the most important tool in the treatment and prevention of bacterial infections and diseases that may arise from therapeutic treatment. However, as Fleming had warned, humanity today is waging a war against bacteria that are resistant to multiple antibacterial agents due to their reckless misuse and abuse [1].

The Gram-negative bacillus *A. baumannii* is a pathogen naturally present in various environments such as water, soil, and human skin, and is a major source of nosocomial infections with known multidrug-resistant strains [2]. Notably, the increased prevalence of multidrug-resistant *A. baumannii* (MRAB) in nosocomial infections, has increased the prevalence of severely ill patients and the antimicrobial resistance rate [3]. Defined as being resistant to three or more antimicrobial classes, including ß-lactams, Aminoglycosides, and Fluoroquinolones, MRAB has also recently become increasingly resistant to carbapenem-type antibacterial agents, which is a serious therapeutic concern [4].

Glycylcycline-class tigecycline and polypeptide colistin (polymyxin E) are currently the most common treatments for MRAB infections [5,6]. Tigecycline is used as a last resort treatment, but antimicrobial resistance owning to efflux pump overexpression is increasing, and tigecycline blood concentrations tend to be low when administered intravenously [7,8]. Moreover, the use of colistin had previously been discontinued due to renal toxicity, and its use for clinical treatment requires appropriate dosing and clinical observation [9].

The combined use of two or more antimicrobial agents is recommended to reduce toxicity and/or because a favorable therapeutic effect cannot be achieved by using a single antimicrobial agent [10,11]. Antimicrobial resistance is increasing in prevalence and diversifying. Consequently, the concept of treatment through natural products that can replace or supplement antimicrobial agents is gaining interest [12]. Probiotics, which have gained attention in recent years, have been confirmed to have a therapeutic effect on various diseases, including chronic inflammatory diseases, and are known to exhibit antimicrobial activity against pathogenic microorganisms through their production of organic acids, bacteriocins, and peptides [13,14]. *Lactobacillus* spp. are classified as “generally regarded as safe” by the United States Food and Drug Administration and are used as a natural antimicrobial agent, Furthermore, *Lactobacillus* spp. have been recognized as an appropriate therapeutic agent that can replace existing antimicrobial agents [15].

This study aimed to identify the synergistic and antagonistic effects of the combined use of tigecycline and colistin with other universal antimicrobial agents (cephalosporin, carbapenem, and quinolone classes) against 22 different strains of MRAB in vitro, as well as the synergistic effect of incorporating a mixed culture of *Lactobacillus* spp. culture extract.

## 2. Materials and Methods

### 2.1. Isolation of Experimental Strains

The Institutional Review Board Deliberations of Dankook University approved this study (IRB No. DKU 202110038, 22 October 2021). The need for informed consent was waived as this study does not use personally identifiable information of any subject.

*Acinetobacter baumannii* strains were isolated from pus, urine, and other specimens tested at the Department of Laboratory Medicine in a university hospital located in Gyeonggi-do, Korea, over a 3-year period from January 2017 to December 2019. The isolated strains were identified using the VITEK 2XL and VITEK-MS microbial identification systems with the GN ID card (bioMerieux, Marcy-l’Etoile, France) [16].

*Lactobacillus* spp. were identified using the VITEK 2XL and VITEK-MS systems from commercially available *Lactobacillus* powder preparations and strains suspected to be *Lactobacillus* from clinically derived specimens, including vaginal and urine samples, tested at the Department of Laboratory Medicine.

### 2.2. Identification of Lactobacillus spp.

A 16S rRNA sequence analysis was performed to accurately identify four strains identified to be *Lactobacillus* spp. via the VITEK 2XL system [17]. After culturing the strains at 37 °C for 24 h in Deman, Rogosa, and Sharpe broth (MRS, BD Difco, Franklin Lakes, NJ, USA), a genomic DNA extraction kit (Thermo Fisher Scientific, Waltham, MA, USA) was used to isolate chromosomal DNA. Universal primers 27F (5′-AGAGTTTGATCCTGGCTCAG-3′, forward primer) and 1492R (5′-GGCTACCTTGTTACGACTT-3′, reverse primer) were used for amplification of the isolated DNA and the analyzed sequences were searched and compared in GENBANK using BLAST from the National Center for Biotechnology Information.

### 2.3. Antimicrobial Susceptibility Test

Antimicrobial susceptibility tests on the isolated *A. baumannii* strains were performed using the AST-N225 card (bioMerieux, Hazelwood, MO, USA) for the VITEK 2XL system to measure the minimum inhibitory concentration (MIC). Tests were performed in accordance with the Clinical and Laboratory Standards Institute guideline M100-S32 [16].

The antimicrobial agents used consisted of the carbapenem (imipenem and meropenem), fluoroquinolone (ciprofloxacin), ß-lactam (piperacillin, piperacillin-tazobactam, ampicillin/sulbactam, ticarcillin-clavulanic acid, ceftazidime, cefotaxime, cefepime, and aztreonam), aminoglycoside (amikacin and gentamicin), tetracycline (tigecycline and minocycline), polymyxins (colistin), and sulfonamide (trimethoprim-sulfamethoxazole) classes of antimicrobial agents.

### 2.4. Multi-Locus Sequence Typing

Multi-locus sequence typing (MLST) analysis was utilized to elucidate gene diversity and structural correlations in the *A. baumannii* strains using seven structural genes from the Oxford housekeeping gene group. Polymerase chain reaction was performed using a solution (50 µL) consisting of 1.25 U of TaKaRa Ex Taq^TM^, 2 × buffer (25 mM TAPS, 50 mM KCl, 2 mM MgCl_2_, 1 mM 2-mercaptoethanol), 0.4 mM of dNTPs mixture, 1 µL of 10 pM primer, and 10 ng of pure isolated chromosomal DNA [18,19]. Sequencing was performed on the resultant amplicons and results were analyzed using the public databases for molecular typing and microbial genome diversity (https://pubmlst.org/ accessed on 14 November 2021) to confirm the final sequence types (STs) of *A. baumannii*.

### 2.5. Checkerboard Test

Twenty-two strains of MRAB (13 × ST191, 4 × ST451, and 5 × ST784 strains), while meropenem, ciprofloxacin, ceftazidime, and tigecycline were used in combination with colistin.

Cation-adjusted Muller–Hinton broth (140 µL; MHB; BD Difco, USA) was added to a 96-well plate (SPL, Pocheon-si, Gyeonggi-do, Republic of Korea), after which, 20 µL of each of the two antimicrobial agents being tested for a combination effect were added at different concentrations. Subsequently, the inoculated solution was incubated with MRAB (37 °C, 24 h) and the final concentration of the bacterial solution in Muller–Hinton broth was brought to 5 × 10^5^ colony forming units (CFU)/mL. Samples were incubated for an additional 24 h at 37 °C and the results were interpreted [20].

Meanwhile, the fractional inhibitory concentration index (FICI) of each antimicrobial combination were calculated using Equation (1). Indices of <0.5, 05–1, 1–4, and >4 were interpreted as synergistic, partially synergistic, no interaction, and antagonistic, respectively [11].
FICI = MIC of antimicrobial agent A in cells using antimicrobial combination therapy/MIC of cells using antimicrobial monotherapy + MIC of antimicrobial agent B in cells using antimicrobial combination therapy/MIC of cells using antimicrobial monotherapy(1)

### 2.6. Time–Kill Assay Test

Time–kill assay tests were performed on combinations of antimicrobial agents that exhibited a synergistic effect in the checkerboard test [21,22]. The strains tested and the combinations of antimicrobial agents were the same as those used in the checkerboard test. The concentration of the inoculated solution was set to 5 × 10^5^ CFU/mL and 20 µL each of the two antimicrobial agents was used.

In addition, culture solutions containing four *Lactobacillus* spp. were used to elucidate their antimicrobial effect using the following procedure: *Lactobacillus* spp. culture solution was incubated on MRS agar (BD Difco, USA) for 24 h, after which, incubated *Lactobacillus* spp. were adjusted using the McFarland standard 4.0. Subsequently, a 0.45 µm syringe membrane filter (Merck Millipore, Burlington, MA, USA) was used to filter out solid components and 20 µL was added to bring the volume of the reaction solution to 200 µL. MRS broth contains sodium acetate, which decreases its pH to inhibit the growth of other bacteria. To differentiate from the low-pH culture solution produced by *Lactobacillus,* brain-heart infusion broth (BD Difco, USA) was used as the dilution solution in all experimental procedures. Subsequently, the mixed broth was incubated at 37 °C for 0, 1, 2, 3, 4, 12, and 24 h, after which, the number of bacterial cells was counted [23,24].

### 2.7. Data Analysis and Statistics

Data analysis was performed using Microsoft Excel (Microsoft, Redmond, WA, USA). For all experimental results, the mean value of three repeated measures was calculated.

## 3. Results

### 3.1. Isolation of Experimental Strains

A total of 33 strains of *A. baumannii* were identified using the VITEK 2XL and VITEK-MS systems, while a total of four *Lactobacillus* spp.—*L. plantarum* and *L. acidophilus* from *Lactobacillus* powder preparation and *L. casei* and *L. crispatus* from human specimens—were also isolated and identified.

### 3.2. Identification of Lactobacillus spp.

For accurate identification of the four *Lactobacillus* spp. using the VITEK 2XL system, 16S rRNA sequence analysis was performed. The strain that was identified as *L. casei* was ultimately determined to be *L. paracasei* based on phylogenetic tree analysis. For the other three strains, the results were consistent with identification using the VITEK 2XL system (Table 1).

### 3.3. Antimicrobial Susceptibility Test

The antimicrobial susceptibility tests on the 33 isolated strains of *A. baumannii* exhibited strong resistance to most carbapenem- and fluoroquinolone-class antimicrobial agents and 100% resistance to most ß-lactam-class agents (piperacillin, piperacillin-tazobactam, ticarcillin-clavulanic acid, ceftazidime, cefotaxime, cefepime, and aztreonam). Three strains (9%) exhibited susceptibility and eight strains (24%) exhibited moderate resistance to ampicillin/sulbactam. A total of 22 strains (67%) and 26 strains (79%) exhibited resistance to the aminoglycoside-class agents amikacin and gentamicin, respectively, with one strain exhibiting moderate resistance. Moreover, 25 strains (76%) exhibited resistance to the sulfonamide-class agent trimethoprim-sulfamethoxazole, while 30 strains (91%) exhibited susceptibility to the tetracycline-class agent tigecycline, with 3 strains (9%) exhibiting moderate resistance. Except for 2 strains (6%) exhibiting moderate resistance, the remaining 31 strains (94%) exhibited susceptibility to minocycline. All strains (100%) exhibited susceptibility to the polymyxins-class agent colistin.

Among the experimental strains, 26 strains (79%) were MRAB, while 7 strains (21%) exhibited susceptibility or moderate resistance to aminoglycoside-class amikacin and gentamicin. Accordingly, these seven strains were selected for further experimentation.

### 3.4. Multi-Locus Sequence Typing

A total of 5 STs were identified from the 33 isolated *A. baumannii strains*—ST191, ST451, ST784, ST229, and ST369. Of them, ST191 was the largest, containing 45% (*n* = 15) of all strains, followed by ST451, ST784, and ST229 with 15% (*n* = 5) each, and ST369 with 9% (*n* = 3; Table 2).

### 3.5. Checkerboard Test

The combination of meropenem and colistin exhibited the strongest overall synergistic effect, with an inhibitory concentration that was four times lower than the inhibitory concentration of meropenem monotherapy (128–256 µg/mL). In particular, the combination of meropenem and colistin exhibited an inhibitory effect, with an MIC (0.5 µg/mL) being four times lower than that of colistin monotherapy; the synergistic effect was also confirmed with a FICI of 0.5. The combination of ceftazidime and colistin exhibited a lower FICI than the combination of meropenem and colistin, but when colistin was combined with ceftazidime, the MIC was 16 times lower than that of colistin monotherapy (1.0 µg/mL) for ST451 and ST784. In addition, a partial synergistic effect was found for ST191, ST451, and ST784 with FICI values of 0.75, 0.56, and 0.56, respectively.

Combination therapy with tigecycline exhibited a partial synergistic effect, while combination therapy with ciprofloxacin exhibited no synergistic effect when combined with colistin (Table 3).

### 3.6. Time–Kill Assay

For ST191, which exhibited the best FICI (0.5), the combination of meropenem (64 µg/mL) and colistin (0.125 µg/mL), and the combination of ciprofloxacin (64 µg/mL) and colistin (0.5 µg/mL), exhibited the largest inhibitory effect when *Lactobacillus* spp. (*n* = 4) culture solution was added (Figure 1). The combination of ceftazidime (32 µg/mL) and colistin (0.5 µg/mL), and the addition of *Lactobacillus* spp. culture solution exhibited a complete inhibitory effect in all experimental groups. However, the combination of tigecycline (2 µg/mL) and colistin (0.25 µg/mL) in treatment groups without *Lactobacillus* spp., with *L. crispus*, and with *L. acidophilus* culture solutions exhibited re-growth of MRAB at 12 h after incubation.

Different combinations of antimicrobial agents exhibited diverse antimicrobial activities against the ST451 and ST784 groups. The combination of ciprofloxacin (64 µg/mL) and colistin (0.5 µg/mL) exhibited an inhibitory response in all experimental groups, including combinations containing *Lactobacillus* sp. (*n* = 4) culture solutions. Conversely, the combination of meropenem (8 µg/mL) and colistin (0.25 µg/mL), and the combination of ceftazidime (64 µg/mL) and colistin (0.25 µg/mL), exhibited no inhibitory effect in all experimental groups.

The results from combination therapy using tigecycline were especially notable. A complete inhibitory effect was found only in the experimental group with the addition of *L. paracasei* culture solution (Table 4).

In all other experimental groups, an inhibitory effect was found up to 4 h after incubation, but re-growth of MRAB at 12 h after incubation was found in the group with no *Lactobacillus* spp., *L. crispatus*, and *L. acidophilus* culture solutions added. Moreover, the re-growth of drug-resistant bacteria that had previously been inhibited was found at 24 h after incubation in the experimental group with the addition of *L. plantarum* (Figure 2).

## 4. Discussion

*Acinetobacter baumannii* is a nosocomial pathogen that can enter and spread throughout the body via various routes, such as through the bloodstream, open wounds, artificial respiratory tract, and urinary tract, especially in immunocompromised patients. The recent increase in drug-resistant strains is an emerging concern for infection treatment [25]. In particular, a high rate of resistance to carbapenem-class antimicrobial agents is being reported [4] and MRAB tends to exhibit susceptibility to only a limited number of antimicrobial agents, such as colistin, tigecycline, and Aminoglycosides [26]. In this study, susceptibility to colistin and tigecycline was found in most strains.

Many studies have recently been conducted on antimicrobial combination therapy and in vitro combination therapy with the addition of natural antimicrobial substances to improve upon existing monotherapy treatments of MRAB infections and the high risk of adverse events associated with such therapy [11,13,26,27].

In the current study, 26 of the 33 strains isolated (79%) were identified to be MRAB, of which, ST191 was the most common type (45%; *n* = 15). In a study by Jun et al. [28] where MLST analyses of 96 strains of carbapenem-resistant *Acinetobacter baumannii*—collected over a 3-year period between 2016 and 2018—were performed, 8, 12, 11, 21, 34, and 10 strains were ST191, ST208, ST229, ST369, ST451, and ST784, respectively, exhibiting a similar ST distribution to the current study.

In the checkerboard test, carbapenem-class meropenem and colistin exhibited the best combination effect (FICI = 0.5). Although a study by Sopirala et al. [11] used different antimicrobial agents, it reported that carbapenem-class antimicrobial agents exhibited an increased antimicrobial effect of carbapenem-class antimicrobial agents when combined with colistin. Moreover, a study by Ju et al. [29] calculated a FICI of 0.03–0.56 when using combination therapy with meropenem and colistin with most FICIs close to 0.5, which are comparable to those of the current study.

In particular, some combinations of antimicrobial agents lowered the concentration of colistin used. The combination of colistin with either meropenem or ceftazidime reduced the amount of colistin used by 4 and 1 -times, respectively, indicating that combination therapies may be able to reduce the burden of renal toxicity associated with colistin. However, further studies are needed on various conditions and dose setting for combining different antimicrobial agents for this to be a viable alternative.

Time–kill assay results demonstrated that *Lactobacillus* spp. culture extracts had an antimicrobial effect on MRAB. The species that exhibited the best antimicrobial effect was *L. paracasei,* which also exhibited the fastest and most sustained antimicrobial activity. *Lactobacillus plantarum* also exhibited a superior antimicrobial effect relative to other species against ST191 and ST451 when combined with tigecycline and colistin. Notably, complete inhibition against ST191 was observed, whereas a partial inhibitory effect against ST451 was demonstrated with inhibition sustained for up to 12 h after incubation, followed by the re-growth of MRAB.

The inhibitory effect of *Lactobacillus* spp. against pathogenic bacteria can be attributed to the production of lactic acid and organic acids, the reduction of entero-adhesion and aggregation by pathogenic bacteria, and the production of antimicrobial substances such as bacteriocins [30]. Especially in vivo, low pH due to lactic acid and organic acids produced by these probiotics and the production of antimicrobial substances, such as bacteriocins, are suspected to directly inhibit the growth and development of pathogenic microorganisms.

## 5. Conclusions

In this study, we demonstrated that some combinations of colistin with another antimicrobial agent lowered the minimum inhibitory concentration of colistin. This suggests that combination therapies may be able to reduce the burden of renal toxicity caused by the use of colistin. Furthermore, we demonstrated that the addition of *Lactobaciilus* spp. to combinations of two antimicrobial exhibited a synergistic antibacterial effect against MRAB.

Based on these results, optimal conditions for the combined use of colistin and other antibacterial agents were established, and the synergistic effect of the combined use of probiotics and antimicrobial agents was confirmed. These findings have the potential to drastically reduce the risk of toxicity by enhancing the antibacterial effect of existing antimicrobials and reducing the amount of colistin required for treatment.

## Figures and Tables

**Figure 1 medicina-59-00947-f001:**
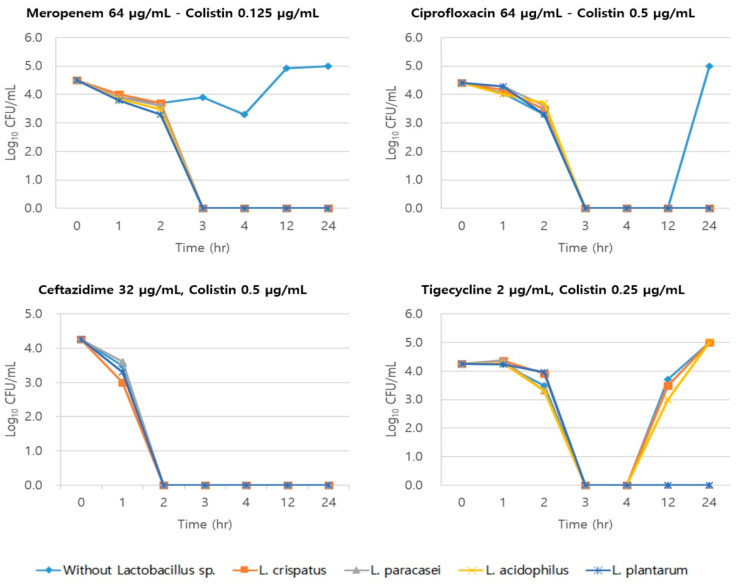
Results of the time–kill assay of MRAB, ST191.

**Figure 2 medicina-59-00947-f002:**
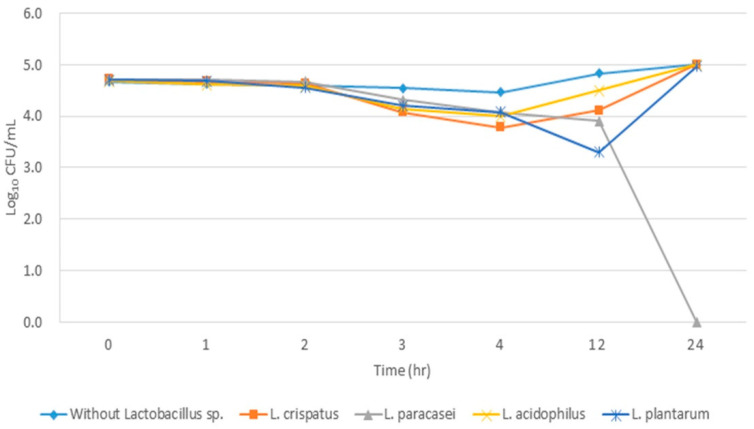
Time–kill assay test curve of tigecycline (1 µg/mL), colistin (0.5 µg/mL), and *Lactobacillus* sp. extract mixture for the MRAB ST451 isolate.

**Table 1 medicina-59-00947-t001:** Result of species identification of probiotics using 16S rRNA sequencing.

Strain Number	Species Identification	Identity (%)	GenBank Accession No.
1	*Lactobacillus plantarum*	99	NR_115605.1
2	*Lactobacillus acidophilus*	99	NR_113638.1
3	*Lactobacillus crispatus*	99	NR_041800.1
4	*Lactobacillus paracasei*	99	AP012541.1

**Table 2 medicina-59-00947-t002:** Distribution of multi-locus STs of 33 *Acinetobacter baumannii* isolates.

Sequence Type	Allelic Profile							No. of Isolates (%)
	*gltA*	*gyrB*	*gdhB*	*recA*	*cpn60*	*gpi*	*rpoD*	
ST191	1	3	3	2	2	94	3	15 (45%)
ST451	1	3	3	2	2	142	3	5 (15%)
ST784	1	3	3	2	2	107	3	5 (15%)
ST229	1	15	2	28	1	107	32	5 (15%)
ST369	1	3	3	2	2	106	3	3 (9%)

ST, sequence type; *gltA*, citrate synthase gene; *gyrB*, DNA gyrase subunit B gene; *gdhB*, NAD-specific glutamate dehydrogenase; *recA*, recombinase A gene; *cpn60*, chaperonin protein 60 genes; *gpi*, glucose-6-phosphate isomerase gene; *rpoD*, RNA polymerase sigma factor gene.

**Table 3 medicina-59-00947-t003:** Results of the checkerboard test of MRAB ST191, ST451, and ST784.

Sequence Type (*n* = 22)	Antibacterial Combination		Single Antibacterial MIC (µg/mL)		Combination Antibacterial MIC (µg/mL)		FIC Index
	A	B	A	B	A	B	
ST191 (*n* = 13)	MEM	COL	256	0.5	64	0.125	0.5
	CAZ	COL	128	1	32	0.5	0.75
	TIG	COL	4	1	0.5	0.5	0.63
	CIP	COL	512	0.5	64	0.5	1.13
ST451 (*n* = 4)	MEM	COL	128	0.5	32	0.125	0.5
	CAZ	COL	128	1	64	0.0625	0.56
	TIG	COL	2	1	0.5	0.5	0.75
	CIP	COL	256	0.5	128	0.25	1.00
ST784 (*n* = 5)	MEM	COL	128	0.5	32	0.125	0.5
	CAZ	COL	128	1	64	0.0625	0.56
	TIG	COL	2	1	0.5	0.5	0.75
	CIP	COL	512	0.5	512	0.25	1.50

A, Meropenem, Ceftazidime, Tigecycline, Ciprofloxacin; B, Colistin; MEM, Meropenem; CAZ, Ceftazidime; TIG, Tigecycline; CIP, Ciprofloxacin; COL, Colistin; MIC, Minimum inhibitory concentration; FIC, Fractional inhibitory concentration.

**Table 4 medicina-59-00947-t004:** Time–kill assay test result table of tigecycline (1 µg/mL), colistin (0.5 µg/mL), and *Lactobacillus* sp. extract mixture for the multi-drug resistant *Acinetobacter baumannii* ST451 isolate.

Species Identification	Number of Bacterial Cells (Log_10_ CFU/mL)						
	Incubation Time (h)						
	0	1	2	3	4	12	24
Without *Lactobacillus sp.*	4.7	4.6	4.6	4.5	4.5	4.8	5.0
*L* *actobacillus crispatus*	4.7	4.7	4.6	4.1	3.8	4.1	5.0
*L* *actobacillus paracasei*	4.7	4.7	4.7	4.3	4.1	3.9	0.0
*L* *actobacillus acidophilus*	4.7	4.6	4.6	4.1	4.0	4.5	5.0
*L* *actobacillus plantarum*	4.7	4.7	4.6	4.2	4.1	3.3	5.0

## Data Availability

The data used to support the findings of this study are available from the corresponding author upon reasonable request.

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
