# Peer review of "Synergistic Effects of a Probiotic Culture Extract and Antimicrobial Combinations against Multidrug-Resistant Acinetobacter baumannii"

_medicina, 2023, doi:10.3390/medicina59050947_

Round 1
Reviewer 1 Report
Lee et al had a study on he effect of combination therapy using conventional
antimicrobial agents that are used for treating drug-resistant bacteria and the additional synergistic effect of four probiotic culture extracts isolated from the human body and Lactobacillus preparations.
The introduction does not provide sufficient background. It is so short. Besides it does not include all relevant references. The references are little and so old. It must be improved.
The research design is not appropriate. The methods need reference(s) and more detail. For example, Isolation of experimental strains section needs reference(s) and more details. About other section it can be true as well.
The method and result section suffer from existing figures or images. I think some figure are needed for method and results sections for more validation and better investigation.
What was the statistical method?? You have some groups. Then, you must use statistical analyzing method as well as post-hoc test.
Conclusion needs to improve. It dose not support the results.
Author Response
" Please see the attachment."

Reviewer 2 Report
Review of “Synergistic effects of a probiotic culture extract and antimicrobial combinations against multi-drug resistant Acinetobacter baumannii” by Ji Hyeon Lee, et al.
The subject of the manuscript is interesting and relevant: Development of pharmacologic and non-pharmacologic therapeutic options to combat emerging antimicrobial resistance in the Acinetobacter baumanii. This nosocomial pathogen is rapidly becoming even more difficult to treat and new strategies are needed to help patients survive infections.
The authors report the susceptibility and sequence types of 33 A baumanii isolates from their institution over a three year period. They then report the antimicrobial efficacy (MICs) and synergy / antagonism of various antibacterial combinations against A baumanii isolates in standard checkerboard assays. Further, they assess the antibacterial efficacy of Lactobacillus extract products by adding them to selected antibiotic combinations in time-kill assays. The methodology is overall sound although there could be more information on how many replicate experiments were done, and a bit more statistical analysis beyond descriptive stats.
The latter experiment (time-kill assays), could be strengthened by characterizing the Lactobacillus extract products further. What active ingredients do the Lactobacillus extract products contain? Controls such as a non-Lactobacillus microbiota extract and a Lactobacillus media (McFarland-MRS broth, etc without bacteria) would be useful to assess whether the Lactobacilli had any effect. What led to the antibacterial effect of the extract and by what mechanism? Could the antibacterial effect be neutralized? If so, how? Perhaps these studies are forthcoming.
The English language aspects of the manuscript could also use some revision. Some of the manuscript, particularly the abstract is difficult to comprehend at first read, and there are grammatical and syntax issues sprinkled throughout that distract the reader. I recommend allowing a dedicated English-language reviewer to proofread it rather than giving line by line corrections herein.
Author Response
"첨부파일을 참조하세요."

Round 2
Reviewer 1 Report
Thanks for the modification. In my opinion, the statistical method is necessary.